Prevalence and seroepidemiology of Haemophilus parasuis in Sichuan province, China

Wang Zhenghao
Zhao Qin
Wei Hailin
Wen Xintian
Cao Sanjie
Huang Xiaobo
Wu Rui
Yan Qigui
Huang Yong
Wen Yiping 100329erga@sina.com
College of Veterinary Medicine, Sichuan Agricultural University , Chengdu , China
Guo Yuming
Electronic publication date: 2017 May 30
Publication date: 2017
Volume: 5
Electronic Location ID: e3379
Received 2017 Feb 9; Accepted 2017 May 5
Copyright: ©2017 Wang et al.
Copyright year: 2017
Copyright holder: Wang et al.
License: This is an open access article distributed under the terms of the Creative Commons Attribution License, which permits unrestricted use, distribution, reproduction and adaptation in any medium and for any purpose provided that it is properly attributed. For attribution, the original author(s), title, publication source (PeerJ) and either DOI or URL of the article must be cited.
License URL: https://creativecommons.org/licenses/by/4.0/

Keywords: Haemophilus parasuis, Seroepidemiology, Serovar

Funding: Haemophflussuis and Porcine Contagious Pleuropneu-monia Prevention Technology Research and Demonstration Project 201303034 This work was supported by the Haemophflussuis and Porcine Contagious Pleuropneumonia Prevention Technology Research and Demonstration Project (201303034), which was financed by public service sector (Agriculture) research project special funds. The funders had no role in study design, data collection and analysis, decision to publish, or preparation of the manuscript.

==============================
Haemophilus parasuis, the causative agent of Glässer’s disease, has been reported as widespread, but little is known about its epidemiology in the Sichuan province of China. The goal of our research is to reveal the prevalence and distribution of H. parasuis in this area. Sampling and isolation were performed across Sichuan; isolates were processed using serotyping multiplex PCR (serotyping-mPCR) and agar gel diffusion (AGD) for confirmation of serovar identity. This study was carried out from January 2014 to May 2016 and 254 H. parasuis field strains were isolated from 576 clinical samples collected from pigs displaying clinical symptoms. The isolation frequency was 44.10%. Statistically very significant differences of infection incidence were found in three age groups (P < 0.01) and different seasons (P < 0.01). Serovars 5 (25.98%) and 4 (23.62%) were the most prevalent, however, non-typeable isolates accounted for nearly 7.87%. In terms of geographical distribution, serovars 5 and 4 were mostly prevalent in west and east Sichuan. The results confirmed that the combined approach was dependable and revealed the diversity and distribution of serovars in Sichuan province, which is vital for efforts aimed at developing vaccine candidates allowing for the prevention or control of H. parasuis outbreaks.

Introduction

Haemophilus parasuis, a member of the family Pasteurellaceae, not only colonises in the upper respiratory tract of healthy pigs during different breeding periods, but also emerges as the aetiological agent of Glässer’s disease in some cases (Oliveira & Pijoan, 2004). This organism incurs significant economic losses within the swine industry. Therefore, efforts have been in studying the epidemiology and pathogenesis of this microbe. So far, 15 serovars and a considerable number of non-typeable isolates have been identified. Various molecular approaches have revealed the diversity in genotypic lineage and have shed light on the heterogeneity of pathogenic H. parasuis serovars (Olvera, Calsamiglia & Aragon, 2006; Del Río et al., 2006; Zhang et al., 2011).

Traditionally, agar gel diffusion (AGD) and indirect hemagglutination (IHA) are used to serotype H. parasuis field isolates (Kielstein & Rapp-Gabrielson, 1992; Del Río, Gutiérrez & Rodríguez Ferri, 2003). In addition, the latter was once proposed as a more discriminatory method for improving the rate of serotyping (Turni & Blackall, 2005). Subsequently, a multiplex PCR was reported for rapid molecular serotyping based on the analysis of variation within the capsule loci (Howell et al., 2015). Each serovar has its own distinctive markers except serovars 5 and 12, for they share the same marker. AGD using specific hyperimmune antisera may be applied to remedy the serotyping issues seen in distinguishing between serovars 5 and 12.

Although serovars 1, 2, 3, 4, 5, 8, 10, 12, 13, 14 and 15 are considered to be virulent, production of cross-protecting vaccines among different serovars has met with little success (Rapp-Gabrielson et al., 1997; Brockmeier et al., 2013). Commercial vaccines for use in swine seldom match the causative serovars and inoculation tends to be ineffective, illustrating the fact the commercial vaccines have, thus far, proven incapable of serovar cross protection in China. Detailed information of morbidity and seroepidemiology can play an important role in prevention when applied in conjuction with research aimed at uncovering potential antigenic markers that are highly conserved across multiple serovars. As no detailed report of H. parasuis infection was available in Sichuan province, China, we investigated the prevalence and seroepidemiology of H. parasuis in Sichuan province from January 2014 to May 2016.

Materials and Methods

Reference strains and hyperimmune antisera

Reference strains of serovars 5 and 12 were purchased from Bacteriology Research Laboratory, Animal Research Institute, Yeerongpilly 4105, Australia. Subsequently, corresponding hyperimmune reference antisera of rabbit source were prepared (Kielstein & Rapp-Gabrielson, 1992).

Collection of samples

From January 2014 to May 2016, a total of 576 clinical samples were collected from 576 weak or moribund pigs showing respiratory distress or arthritis. The survey focused on 103 intensive swine farms in the Sichuan province. The farms covered 28 counties of 13 cities of Sichuan province. Incidentally, the province was divided into two parts artificially by climate rather than vertical division. The annual average temperature of the northeast region was a bit higher than that of the southwest region in Sichuan province.

All pigs were necropsied humanely. Lungs, heart blood, joints and brain were derived from pigs showing signs of acute infection or septicemia, while lungs, pericardium liquid, pleural effusion, seroperitoneum and joint fluid as well as fibrinous membrane were collected from pigs displaying relatively mild symptoms (Turni & Blackall, 2007). At least two tissues or exudates per pig were analyzed in the present study. Different tissues or exudates from the same pig were considered to be one sample. Additionally, all samples were classified by age groups, seasons and regions.

Isolation and identification

Samples were inoculated on tryptic soy agar (TSA; Becton, Dickinson and Company, Franklin Lakes NJ, USA) with a final concentration of 5% newborn calf serum and 1 µg/ml of nicotinamide adenine dinucleotide (NAD; Roche, Basel, Switzerland). The plates were incubated for at least 24 h at 37 °C under aerobic conditions. Translucent colonies of 1 mm in diameter were further passaged. Subsequently, Gram staining was carried out and culture with a typical morphology of H. parasuis such as threadless and medium or short rod was confirmed by PCR as described elsewhere (Oliveira, Galina & Pijoan, 2001). Moreover, strains isolated from different tissues of a pig were counted as one isolate.

Serotyping

First, all the isolates were serotyped by multiplex PCR using molecular markers and reaction procedure as previously described (Howell et al., 2015). As noted previously, AGD was performed to differentiate between serovars 5 and 12. Pure cultures of the isolates with the marker of serovars 5 and 12 were grown in tryptic soy broth (TSB; Becton, Dickinson and Company, Franklin Lakes, NJ, USA) and cells were harvested by centrifugation at under 8,000× g for 5 min, and then resuspended in phosphate buffered saline (pH: 7.4); cell concentration was adjusted (OD600 = 1.2 ± 0.1). The cells was heated at 121 °C for 2 h and followed by a centrifugation under 6,000×g for 5 min and supernatant was collected as heat-stable antigen. Using the hyperimmune antisera against serovars 5 and 12 earlier prepared, AGD was performed with 3% agarose by using heat-stable antigens of standard strains as positive controls, ultrapure water as a blank control and Actinobacillus pleuropneumoniae as a negative control. A single immune line between antiserum holes and heat-stable antigen holes indicated positives. Strains of the same serovar from a single farm were counted as one isolate.

Statistical analysis

Data was statistically analyzed by the procedure of IBM SPSS Statistics 19.0, and differences between age groups, seasons and regions were performed by Chi-square analysis. All tests were 2-sided and P < 0.01 was defined as a very significant difference while P < 0.05 was defined as a significant difference.

Results

Bacterial strains

In the present investigation, varying degrees of respiratory symptom and classic Glässer’s disease were found in herds of the intensive swine farms (Fig. 1). A total of 254 field H. parasuis isolates were confirmed via viewing culture morphology characteristics and 16S rRNA PCR results, with the isolation rate of 44.10%. The infection incidence of H. parasuis was 45.98% and 43.77% in 2014 and 2015, respectively, and 41.10% was measured in the first five months of 2016, showing no significant differences (χ2 = 0.531, P > 0.05). However, infection incidence varied with age, season and region In pigs of 3-8 weeks, the general infection incidence was 60.80%, which was much higher than that of pigs of less than three weeks or over eight weeks. Statistically very significant differences were found in three age groups as whole (χ2 = 108.209, P < 0.01).

Moreover, infection incidence in winter and spring were much higher than the other two seasons, and statistical differences were very significant (χ2 = 27.854, P < 0.01) which indicated winter and spring were the major epidemic seasons of H. parasuis in Sichuan province. However, no significant differences were seen in two regions as whole (χ2 = 3.031, P > 0.05) (Table 1).

Figure 1 Gross lesions in H. parasuis infected pigs.

(A) Fibrinous pericarditis. (B) Fibrinous pleuritis.

Table 1 Haemophilus parasuis infections in Sichuan province, China (January, 2014–May, 2016).

Year	Total	Age groups (weeks)	Seasons	Regions	
		<3	3–8	>8	Winter	Spring	Summer	Autumn	East	West	
2014	80/174*	2/24	65/104	13/46	29/61	45/86	6/19	0/8	20/45	60/129	
2015	144/329	3/51	120/186	21/92	67/136	52/93	7/47	18/53	32/83	112/246	
2016	30/73	1/7	29/62	0/4	9/25	21/48	/	/	0/10	30/63	
Total	254/576	6/82	214/352	34/142	105/222	118/227	13/66	18/61	52/138	202/438	
Notes.

* A/B, A represents number of strains, and B represents number of samples.

mPCR and AGD for serotyping

Follow-up serotype-mPCR and complementary AGD examination classified them into 11 serovars, which showed a considerable serovar diversity (Fig. 2; Fig. 3). Serovar 5 (66 of 254, 25.98%) and serovar 4 (60 of 254, 23.62%) were most prevalent. However, serovars 3, 6, 10 and 15 were not found. The typeable and non-typeable isolates were 234 (92.13%) and 20 (7.87%) out of total 254 respectively.

Figure 2 mPCR of serotyping for H. parasuis field isolates and standard strains.

(A) M: DL 2000 DNA Marker (Takara); C: blank control; S1, S2, S4, S5, S7, S8 and S9: standard strains of serovars 1, 2, 4, 5, 7, 8 and 98;FS1, FS2, FS4, FS5, FS7, FS8 and FS9: field isolates AB14011, GA14051, YA14041, GA14032, LS14031, GA15013, NC14041 (There were no field isolates of serovars 3, 6, 10 or 15 obtained in the investigation). (B) M: DL 2000 DNA Marker (Takara, Shiga, Japan); S11, S12, S13, S14, S3, S6, S10 and S15: standard strains of serovars 11, 12, 13, 14, 3,6,10 and 15; FS11, FS12, FS13 and FS14: field isolates CD14021, LS15044, NJ14021, MY14021; N1: non-typeable strain; N2: A. pleuropneumoniae; N3: Streptococcus suis.

Figure 3 Agar gel diffusion (AGD) of serotyping for H. parasuis field isolates.

(A) S1: hyperimmune antiserum of serovar 5; F1: H. parasuis field isolates GA14032; F2: H. parasuis field isolates CD14041; P1 and P2: positive control of reference strain for serovar 5; N1: negative control of A. pleuropneumoniae; N2: blank control. (B) S2: hyperimmune antiserum of serovar 12; F3: H. parasuis field isolates CD14051; F4: H. parasuis field isolates LS15044; P3 and P4: positive control of reference strain for serovar 12; N3: negative control of A. pleuropneumoniae; N4: blank control.

Distribution and tendency of serovars in Sichuan province

In the present study, geographical distribution of serovars indicated that serovars 5 and 4 were the most prevalent in Sichuan province (Fig. 4A). In addition, serovars 1, 7 and 12 were only found in west Sichuan and serovar 8 was only found in east Sichuan (Fig. 4A). On the other hand, serovars 4 and 5 were prevalent in both 2014 and 2015, however, serovar 7 was the lead in prevalence for the first five months of 2016 (Fig. 4B).

Figure 4 Distribution and variation of different serovars of H .parasuis in Sichuan province, China.

(A) Distribution of different serovars of H. parasuis in east and west Sichuan. (B) Variation of different serovars of H. parasuis during a two year five months survey period.

Discussion

Isolation and identification of H. parasuis is the “gold standard” for diagnosis, but it requires that the samples to be collected from acute or typical infection cases without antimicrobial treatment as possible. Furthermore, rigorous nutritional demands and fragility of H. parasuis slow down the culture process and increase the probability of false negatives (Del Río et al., 2003). The isolation rate of previous studies ranged from 0.99% to 21% (Oliveira, Blackall & Pijoan, 2003; Fablet et al., 2012; Zhang et al., 2012). In the present study, the isolation rate was 44.10% in the Sichuan province (Table 1), and revealed a mark differences in prevalence in different areas. Differences of infection incidence among different age groups were in accordance with previous reports, which reconfirmed that H. parasuis infection was a big threat to piglets. In addition, an apparently increased infection rate in winter and spring was due mainly to the sudden change of temperature and poor ventilation.

Before 2015, studies on epidemiology of H. parasuis usually depended on traditional approaches for serotyping. In 2012, 112 (out of 536 pigs) H. parasuis strains were isolated in southern China, and combination of AGD and IHA revealed that serovars 5 and 4 were the most prevalent serovars, whereas non-typeable strains accounted for nearly 20% (Zhang et al., 2012). In Northern Italy, serovars 4, 13 and 5 were demonstrated as the top three serovars in epidemic H. parasuis strains using AGD, however, non-typeable strains accounted for 27.3% (Luppi et al., 2013). The simplex method is always subjected to susceptibility and improper preparations of hyperimmune antisera. By the combination of mPCR and complementary AGD, the profile of prevalent serovars in the Sichuan province has been revealed in the present study (Fig. 4), which confirmed that the combined approach proposed here is efficient and reliable. Considering the existence of non-typeable strains, the most convincing hypothesis is that there is deletion or insertion or mutation in serotype specific genes on the capsule loci. Secondly, there are differences between phenotypic and genotypic serotype in certain bacteria due to other alterations (Gentle et al., 2016). According to the previous report, the two in silico-nontypeable isolates can be differentiate by mPCR which indicated a non-absolute concordance between mPCR and in silico serotype analysis. We also found that some strains cannot be identified by mPCR, but the in silico serotype analysis is able to differrntiate them (Howell et al., 2015). Thus, more efforts have to be made to elucidate the differences. Moreover, some serovars previously believed to be nonvirulent or hypervirulent were also isolated with a frequency of 18.89%, and the discrepancy is due, to some extent, to the immune capacities, feeding conditions and supervision, and so forth (Kielstein & Rapp-Gabrielson, 1992; Brockmeier et al., 2013; Yu et al., 2014).

There are four bivalent and one monovalent inactivated vaccines for H. parasuis authorized in mainland China; however, desired protection are seldom obtained on account of the poor cross protection among these endemic serovars. Therefore, vaccines corresponding to specific serovars fits the region are optimal (Yu et al., 2014). However, no recent seroepidemiology reports were accessible for the Sichuan province and the prevalent serovars may change over time. The serovar profile in the present investigation was partly consistent with previous reports in China mainland, however, following serovars 4 and 5 were serovars 7, 1 and 2 rather than serovars 14, 13 and 12 as previously reported (Cai et al., 2005; Zhang et al., 2012; Chen et al., 2015). Furthermore, despite the most two epidemic serovars being the same in the two parts of the Sichuan province, the secondary serovars may interfere in the control of H. parasuis infection. The present study revealed the seroepidemiology of H. parasuis in Sichuan province, China. Thus, existing vaccines and vaccine candidate strains should be taken into consideration to control the disease. To our knowledge, several vaccine manufacturers and laboratories were considering more appropriate adjuvant candidates to reduce the irritation generated by inactivated vaccines. Furthermore, genetically engineered and attenuated vaccines against virulent strains of different serovars might be promising (Oliveira, Blackall & Pijoan, 2003). Although the prevalent serovars in the present study in a given area may change over a prolonged time span, throughout a two and a half year period, we believe that the serovars percentage profiles remained stable. Subsequent investigation should be carried out on annual basis to understand the variation tendency in seroepidemiology of H. parasuis in Sichuan province.

Conclusion

This study confirmed that there was a high infection rate of H. parasuis in Sichuan province from January 2014 to May 2016. The present research revealed that H. parasuis incidence was the highest in nursery pigs and infection rates in winter and spring were much higher than during the other two seasons. In addition, it is also demonstrated serovars 4 and 5 were the most prevalent in the Sichuan province, and mPCR combined with AGD was dependable for H. parasuis serotyping.

Supplemental Information

Supplemental Information 1 Raw data for strains collection information

Every strain was numbered and serotyped, and the location and date of the isolation is stated (eg: “NC14041” means the strain was the first isolated in April 2014 in Nanchong City).

Click here for additional data file.

We thank the collaborating producers for facilitating the study on their farms, and thank the members of the Swine Disease Center of the College of Veterinary Medicine for their underlying contributions. We especially thank Prof. Yung-Fu Chang of Cornell University for the critical reading and editing of this manuscript.

Additional Information and Declarations

Competing Interests

Author Contributions

Data Availability

The authors declare there are no competing interests.

Zhenghao Wang conceived and designed the experiments, performed the experiments, analyzed the data, wrote the paper.

Qin Zhao and Hailin Wei performed the experiments, prepared figures and/or tables.

Xintian Wen and Sanjie Cao contributed reagents/materials/analysis tools.

Xiaobo Huang, Rui Wu, Qigui Yan and Yong Huang reviewed drafts of the paper.

Yiping Wen analyzed the data, reviewed drafts of the paper.

The following information was supplied regarding data availability:

Raw data can be found in the Supplemental Information.

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
