# Peer review of "Prevalence and seroepidemiology of Haemophilus parasuis in Sichuan province, China"

_PeerJ, doi:10.7717/peerj.3379_

## Round 0.1 · original submission · Major Revisions

Thank you for your manuscript which has been reviewed by the reviewers. Overall, this paper is interesting. But the English should be improved by a native English speaker before resubmission. Based on the comments and recommendation of the reviewers, this manuscript is likely to become acceptable for publication in PeerJ if adequately revised to address all of the review comments.

Reviewer 1 ·

Basic reporting

English can be improved. Some suggestions are included in general comments. Raw data in Tables S1, S2 and S3 are redundant with table 1.

Experimental design

No additional comments that those included in general comments

Validity of the findings

The results can be improved if serotyping of serovar 12 is also performed, not only serovar 5. In addition, the number of samples used should be clarify.

Additional comments

This study provides information on the prevalence of Glässer’s disease and the serovars involved in the clinical cases in Sichuan (China), which can be of local interest.

Line 33. Consider changing “prepotent” to “prevalent” (also in other parts of the manuscript, such as line 170) and “compound” to “combined”
Line 45. It is not clear what the authors mean by “conventional”
Line 52, and 97. Usually, when a reference has 3 authors only the first one followed by et al is indicated.
Line 54. Consider including Del Río et al 2003 J Clin Microbiol, as the first report using IHA for H. parasuis
Line 56. The proper reference for this sentences may be Turni and Blakall 2005.
Line 57. Consider changing “performed” to “described” or “reported”
Line 63. Refernce Susan et al is not correct. It should be Brockmeier et al. In fact on the refence list the same reference is included as Susan et al and Brockmeier et al. (Susan is the first name, not the last name of the author).
Lines 66-67. Sentence “Considering no year-spanning report of H. parasuis infection was approachable in Sichuan province..” is not clear.
Section “Reference strains and hyperimmune antiserum”. First, it should be “antisera”, in plural. But more importantly, in this study only antiserum against serovar 5 is used, therefore only the description of those should be included in this section. The rest of antisera are not used in current study. Serotyping of serovar 12 should be also performed.
Section “Collection of samples”. Authors should clarify the number of samples per pig that were used in the study. Line 77 (“…total of 576 clinical samples of 576 weak or moribund pigs..”) indicates that 1 sample per pig was studied while later, lines 97-98 (“…strains from different tissues of the same pig…”) indicate that more than 1 sample per pig were analyzed.
Line 78. Consider changing “arthrocele” to “arthritis”
Lines 83-86. Samples should be taken from the organs with lesions. Line 83 “Lungs, heart blood, infectious joints and brain were derived from pigs…”, eliminate “infectious” or change to “infected” or “affected”. What is fiberours membrane?
Line 91. Eliminate “respectively”
Line 95. What do the authors mean by “rhabditiform”? This term is usually used for larva
Line 97-98. How did the author choose one isolate over the rest? The best way to determine if several isolates are identical is to use a genotyping method. For H. parasuis, ERIC-PCR has been successfully used to easily differentiate strains.
Lines 101-113. Only serotyping with anti-serovar 5 serum was performed with isolates giving positive results in the PCR for serovars 5/12. Serotyping with anti-serovar 12 serum should be also performed, especially with those strains yielding negative results for serovar 5 in GD.
Line 116. What is “interesting measures”? Please, specify.
Line 122. “Herds” instead of “herbs”
Figure 1. Descriptions may be changed to fibrinous Pericarditis and fibrinous pleuritis
Line 127. What is the meaning of pathogenetic pigs?
Line 128. Indicate the ages in the text. As it is now, you don’t see the ages until consultation of the table.
Lines 129-133. The value of P can not be 0.000. You can indicate “lower than” e.g. P<0.001
Line 138. Check the number of serovars. It seems that it should be 11 instead of 12.
Lines 139-144. The data in the text is identical to that found in Figure 4. Authors can consider eliminating the figure or eliminating the data in the text and referring to the figure.
Line 170. I don’t think that you can say that it is a “novel profile”, since classically serovar 4 and 5 have been the most prevalent, as it was found also here.
Lines 196-197. The authors can analyze the variations during 2014 and 2015 to see if they can observe already a tendency during time and include it in the manuscript.
Tables S1, S2 and S3 present the same information than Table 1, and in fact all the information from those 3 tables (S1, S2 and S3) is included in Table 1. Tables S1, S2 and S3 are not necessary.

Reviewer 2 ·

Basic reporting

Poor English language used throughout. This paper requires a lot of work to ensure that it can be interpreted properly. I would recommend having a native English speaker help with the correction of the manuscript.
E.g. line 122 herbs used when I presume they mean herds.
E.g. line 148
E.g. Line 177-178

At least one reference is incorrect – Susan et al. 2013 is in fact the first name rather than the surname, which should be Brockmeier. Please check all references thoroughly.

Experimental design

Statistical tests used to assess HPS infection rates by age group and by season.
mPCR and GD serotyping of HPS to report the prevalence of serovars in the province in recent years.
It is a very basic experimental design but does provide some new information on the prevalence of HPS in this area of China.

Validity of the findings

From the Bacterial strains paragraph: I am unsure of the importance of the statistical tests for levels of infection per age group, or season when they are predominantly observations on the patterns of infection. This section could be reduced to a more readable format.
Data is straight forward enough but the english language makes it difficult to asses the conclusions any further than the prevalence of serovars.

Additional comments

The paper needs a substantial overhaul on the language and grammar. The surveillance of serovars in a province or country is important information to have for vaccination and planning future preventative measures to reduce disease. However, these are not discussed in detail within the paper.

---

## Round 0.2 · Minor Revisions

Please make a last effort to improve the English language in the manuscript before Acceptance.

## Staff Note: We suggest you seek the help of a native speaker to revise any awkward language phrasings ##

Reviewer 1 ·

Basic reporting

Correct

Experimental design

Correct, and improved with modification on defining samples

Validity of the findings

Correct

Additional comments

In general the authors have followed the recommendations of the reviewers and they have improved the manuscript. My only comment now is that some of the new sentences should be edited to improve the English.

---

## Round 0.3 · accepted · Accept

We hope you will choose PeerJ again for your future work.